# Temporal Analysis of Embryonic Epidermal Morphogenesis in *Caenorhabditis elegans*

**DOI:** 10.3390/ijms262110802

**Published:** 2025-11-06

**Authors:** Fangzheng Li, Peiyue Li, Mao Onishi, Law King Chuen, Yukihiko Kubota, Masahiro Ito

**Affiliations:** Information Biology Laboratory, Department of Bioinformatics, College of Life Sciences, Ritsumeikan University, Kusatsu 525-8577, Japan; sj0073hx@ed.ritsumei.ac.jp (F.L.); lipeiyuesdnbiz@gmail.com (P.L.); onishi@nibb.ac.jp (M.O.); kingchlaw@gmail.com (L.K.C.); yukubota@fc.ritsumei.ac.jp (Y.K.)

**Keywords:** temporal analysis, deep learning, *Caenorhabditis elegans*, epidermal morphogenesis

## Abstract

The development of epidermis plays a central role in driving the morphogenesis of the *Caenorhabditis elegans* embryo. However, current research on epidermal morphogenesis focuses disproportionately on overt phenotypic abnormalities, potentially overlooking the crucial role of developmental timing. In this study, we developed a modular two-step deep learning-based image analysis pipeline. First, we used ResU-Net to extract completely developed embryos and suppress noise; second, ResNet was used to predict the corresponding embryonic stage. The predicted probabilities and their corresponding embryonic time points were subsequently utilized to construct a developmental timeline. Combining this pipeline with differential interference contrast time-lapse microscopy, we dynamically tracked the timeline of epidermal morphogenesis in RNAi-treated embryos (*ajm-1*, *tes-1*, *leo-1*) and mutant embryos (*clk-1*). By statistically comparing the duration of each embryonic stage, our approach enabled the detection of stage-specific developmental timing without relying on overt phenotypic abnormalities or fluorescent markers, successfully recapitulating and extending the known roles of these genes from a temporal perspective. Our work underscores the importance of incorporating developmental timing into morphogenetic analysis, offering a novel framework for revealing subtle developmental processes, deepening the understanding of morphogenetic dynamics, and bridging the methodological gap in *C. elegans* embryology.

## 1. Introduction

The development of the epidermis plays a central role in driving the morphogenesis of the *Caenorhabditis elegans* (*C. elegans*) embryo. Epidermal morphogenesis is a complex and dynamic process that requires the collaboration of multiple features such as mechanical forces and signaling [1]. The epidermis is the outer cellular layer of an organism and plays an important role in embryogenesis as this tissue regulates the shape of the animal during the developmental process [2]. Understanding this process is crucial for uncovering the mechanisms of morphogenesis and has significant implications for studying developmental diseases and related gene functions [3]. *C. elegans* is a useful model to study morphogenesis and provides insights into various human diseases due to its fixed cell number, stereotypical cell division pattern, rapid developmental process, and the ease and economy with which genetic experiments such as RNA interference (RNAi) can be performed [2,4,5,6,7].

Epidermal morphogenesis involves complex cell-to-cell interactions and coordination between multiple genes. The intricate process of epidermal development in embryos is often described by classifying it into distinct phenotypic stages, which provide a valuable framework for the analysis of phenotypic embryonic images (Figure 1A). Several influential studies have reported the critical role of epidermal morphogenesis using defective embryonic images to elucidate aspects such as epidermal cell fate junctions, interactions, and regulatory mechanisms [8,9,10,11,12]. Incorporating a temporal perspective is essential in developmental studies of epidermal morphogenesis, as it provides insights into crucial developmental genes that may not exhibit an obvious or severe phenotype. A pioneering study successfully used RNA-sequencing analysis to explore linkages in developmental timing and provide comprehensive insights into temporal and spatial dynamics [13]. Recently, a novel, large-scale study used RNAi and four-dimensional imaging of specialized strains tagged with fluorescent proteins to systematically characterize embryonic development [14]. However, current image-based research methods focus too heavily on defective phenotypes. Although certain genes are involved in epidermal development, their inactivation does not cause obvious morphological abnormalities, and the epidermal developmental process can still be completed. Instead, the progression through specific embryonic stages is prolonged (Figure 1B). This often requires complex experimental designs aimed at converting developmental delays into observable defects. One such method entails combining weak-allele mutant strains with RNAi to induce more severe developmental defects than can be observed with single mutants. This highlights the need for a novel approach emphasizing the developmental timing and addressing the omissions in traditional approaches during the epidermal development process.

Recent advancements in deep learning-based image analysis have led to substantial breakthroughs across various research fields, particularly in medical imaging, where automated analysis enables more accurate and efficient diagnoses [15,16,17,18,19]. This method has also transformed the approach to developmental biology, expanding our understanding of morphogenesis, embryogenesis, behavior, and even aging in *C. elegans*. For instance, image segmentation models have been used to analyze the “morphodynamics” of early embryos [20], and object detection models have characterized multiple features of *C. elegans* in microfluidic devices, including size, movement, speed, and fluorescence [21], apart from aiding the study of worm behavior [22]. Furthermore, image similarity analysis models have been applied to identify cell divisions in early embryos automatically [23], and classification models have been employed to study lifespan [24]. However, most studies have focused on early embryos, tracking cell lineage or behavior, while studies focusing on epidermal morphogenesis are lacking.

Other studies have aimed to precisely characterize *C. elegans* embryonic developmental timing primarily using cell lineage tracking tools, such as the nucleus segmentation and tracing tools, StarryNite and AceTree [25,26], or the membranes automated segmentation and cell lineage tracing pipeline CShaper [27]. While these tools allow researchers to explore embryonic development at the cellular level through annotated image stacks, complementary methods that enable the analysis of developmental timing at the morphogenetic level during the later embryonic stages are still limited.

This study introduces a deep learning-based image analysis approach, offering a more direct and temporally focused methodology compared to traditional methods. This approach is particularly capable of highlighting cases where the effects on development are not severe enough to cause detectable changes in the phenotype but are manifested as developmental delays. Time-lapse differential interference contrast (DIC) microscopy was used to monitor the developmental timing of each stage of epidermal morphogenesis in *C. elegans*. Our approach combined two classic deep learning models: ResU-Net [28,29] was used for noise reduction, thereby facilitating focus on embryonic regions, and ResNet [30] was employed to predict embryonic stages and generate a developmental timeline to facilitate real-time-tracking of the process. Our findings indicate that in the absence of apparent phenotypic defects, temporally assessing embryogenesis presents an effective approach to detect subtle developmental changes. Additionally, shifting the focus from visible phenotypic defects to developmental timing provides a platform for identifying genes that contribute to morphogenesis.

## 2. Results

### 2.1. Embryo Segmentation and Classification Using ResU-Net and ResNet

#### 2.1.1. Image Segmentation Using ResU-Net

ResU-Net was incorporated into the first step of our pipeline to generate pixel-accurate masks, reducing noise from *E. coli*, worm tissue fluid, or other embryos in the image frame.

Intact embryos in the images were successfully segmented and processed, regardless of the embryo orientation. Of the 765 images, 80.0% were collected for training and 20.0% were used for validation. The ResU-Net model achieved a recall (sensitivity) of 99.1% and a specificity of 98.0%, indicating its reliability to distinguish between segmented and non-segmented areas. The overall accuracy reached 98.5% with a precision (positive predictive value) of 97.2% and an F1-score of 98.1%. Furthermore, the Intersection over Union (IoU) was calculated to be 96.4%, reflecting the model’s effectiveness in accurately localizing relevant regions (Table 1, Figure 2A,B).

The model performed well in the actual timeline experiment (Figure 1B), and maintained accuracy in images with substantial noise around the embryo (Figure 2C).

#### 2.1.2. Image Classification Using ResNet

ResNet was employed to predict the embryo stages from the time-lapse images denoised by ResU-Net. Seventy percent of each label was randomly selected as training data, and the remaining 30.0% for validation. The ResNet model achieved recall (sensitivity) of 96.9%, precision (positive predictive value) of 96.9%, and F1-score of 96.8%. The overall classification accuracy reached 96.86% (Table 2, Figure 3A). Five-fold cross-validation was used to assess robustness, achieving an average accuracy of 94.3% ± 0.4% (see Appendix A).

Gradient-weighted class activation mapping (Grad-CAM) heatmap analysis confirmed that the model captured the key features of each embryonic stage, particularly of the outline at the embryo’s boundary (Figure 3B). Uniform Manifold Approximation and Projection (UMAP) visualization demonstrated that the model effectively separated the different embryonic stages in the feature space. Notably, the spatial arrangement of the stages was not random in UMAP projection but rather followed a trajectory consistent with *C. elegans* embryonic development (Figure 3C).

### 2.2. Analysis of Temporal Prediction Accuracy in RNAi Time-Lapse Data

To construct the timeline of embryonic epidermal development, denoised time-lapse images were used as input for ResNet. The predicted probabilities for each developmental stage were then plotted over time, with each stage visualized using a distinct color, to show the dynamic progression.

Given that worm epidermal development follows a fixed developmental process, any insertion of differently colored points within a continuous period of a single color in the predicted timeline was considered a misclassification (Figure 4A). In this study, among the 16 time-lapse datasets in the *control(RNAi)* group, only 1 misclassified image was found out of a total of 681 images. In the RNAi group, 7 misclassified images were identified among 1465 images from 30 time-lapse datasets. Moreover, neither the control group nor the RNAi group showed continuous insertion of differently colored points within a single stage’s duration (i.e., no consecutive misclassification was detected) (Table 3). These misclassified points were considered as the corresponding correct embryonic stage when calculating the duration of each stage in the subsequent analysis.

Additionally, epidermal development is a continuous process, with a transition period between two adjacent stages. Our current method revealed two pattern types during the transition period. In the first type, oscillations were observed between the two embryonic stages (Figure 4B). In the second type, two stages were mixed with a 40–60% probability (Figure 4C).

### 2.3. Analysis of Embryonic Stage Durations in control(RNAi) Time-Lapse Data

To calculate the time required for each stage, the misclassification images were manually corrected based on their positions relative to the preceding and subsequent embryonic stages. Subsequently, the duration of each embryonic stage was calculated. The average durations for the dorsal intercalation, ventral enclosure, rotation, 1.5-fold, and 2-fold were 53.74 ± 1.85 min, 28.43 ± 2.02 min, 15.93 ± 1.38 min, 20.93 ± 1.04 min, and 10.62 ± 0.89 min, respectively. Additionally, the duration of all transition periods from dorsal intercalation to the 2-fold stage was mostly stable at around 5 min (Figure 5, Table 4).

### 2.4. Application in RNAi Knockdown Time-Lapse Data

To assess the applicability of our approach to gene function analysis, we used our pipeline to predict embryonic stages of RNAi animals and analyzed performance of timeline predictions. We selected three genes, *leo-1*, *ajm-1*, and *tes-1* to test our model. LEO-1, a component of PAF1 complex (PAF1C), is involved in cell localization and cell shape [31]. In contrast, AJM-1, localized at the epidermal apical junctions, is involved in the elongation stages of epidermal development [32]. TES-1 is involved in epidermal elongation from the 2-fold stage through its interaction with the cell junction protein, HMP-1 [33].

Ten time-lapse datasets were analyzed for each RNAi-knockdown embryo to characterize gene function. In total, 541, 467, and 457 images were obtained from *leo-1(RNAi)*, *ajm-1(RNAi)*, and *tes-1(RNAi)* animals, respectively.

Next, after manually correcting the misclassifications, the average duration of each RNAi-treated embryonic stage was calculated to evaluate the potential of the proposed approach for temporal gene function analysis. Additionally, owing to the effects of gene knockdown on embryos, a few instances of highly unstable predicted timelines were observed within the RNAi group (the calculation example of the duration of each embryonic stage in some unstable timelines; Appendix A).

The results for the average duration of each RNAi-treated embryonic stage are listed in Table 4. Compared to the *control(RNAi)*, the *leo-1(RNAi)* exhibited marked developmental delays in the dorsal intercalation stage (delayed by 20.0%; *p* < 0.05), ventral enclosure stage (delayed by 56.5%; *p* < 0.001), and rotation stage (delayed by 44.1%; *p* < 0.01) (Figure 6A–C). In contrast, developmental delay was not observed in the 1.5-fold and 2-fold stages. This marked developmental delay of the early epidermal stages is indicative of LEO-1 as a component of PAF1C and its involvement in early epidermal development.

Compared to the *control(RNAi)*, the *ajm-1(RNAi)* exhibited marked developmental delays in the elongation period: 1.5-fold stage (delayed by 31.4%; *p* < 0.01) and 2-fold stage (delayed by 50.7%; *p* < 0.01) (Figure 6D,E). In contrast, developmental delays were not observed in early epidermal developmental stages. These results are consistent with a previous report stating that AJM-1 is required for the correct rate and completion of elongation in the *C. elegans* embryo.

Compared to the *control(RNAi)*, the *tes-1(RNAi)* exhibited marked developmental delays only in the 2-fold stage (delayed by 93.1%, *p* < 0.0001) (Figure 6E). This result confirms a previous report documenting that TES-1 is recruited to junctions in cells that generate sufficient tension to elongate to the 2-fold stage.

Considering that manual correction may introduce human bias, we also calculated the average duration without correcting the misclassified images as a reference. The results showed that, in the absence of manual correction, only the *ajm-1(RNAi)* group at the 2-fold stage exhibited a change in significance level from *p* < 0.01 to *p* < 0.05, while the results of other stages remained unchanged (see Appendix A)

We also analyzed and compared the duration of all transition periods from dorsal intercalation to the 2-fold stage within the generated timeline but did not observe any significant developmental delay.

### 2.5. Expanding Analysis of Temporal Prediction Accuracy in Developmentally Slow clk-1(e2519) Mutant Embryos

To further extend the applicability of this approach to genetic mutants, we examined the developmentally slow mutant *clk-1(e2519)*. In the wild-type (WT) group, two misclassified images were identified out of a total of 383 images. However, in the *clk-1(e2519)* group, 23 misclassified images were identified out of a total of 1000 images (Table 5), which was relatively high compared to the RNAi-treated groups. Furthermore, we compared the misclassification stages between WT and *clk-1* mutant timelines. Since more misclassifications occurred during the late embryonic stages, we focused on the 1.5-fold and 2-fold when comparing the WT and *clk-1* mutants. We noticed that while misclassifications did not occur in WT, it occurred 9/23 (39.1%) at the 1.5-fold stage and 1/23 (4.3%) at 2-fold stage in the *clk-1* mutant.

Inspection of the time-lapse series revealed that *clk-1* embryos exhibited pronounced muscle-twitching at the 1.5-fold stage, occurring earlier than in the WT embryos, where pronounced muscle-twitching begins post the 2-fold stage (see Appendix A). Further interpretability analysis using Grad-CAM revealed that such pronounced muscle-twitching, combined with the slow elongation, induced morphological change, which occasionally led the classifier to misinterpret dorsal curvature as the ventral side (see Appendix A).

Both the WT and *clk-1* groups exhibited one continuous misclassification each during the dorsal intercalation period. (Table 5, Appendix A). To minimize human bias, only isolated misclassifications were manually corrected, durations of continuous misclassification were excluded from the statistical analysis. As a result, the number timelines analyzed for dorsal intercalation was 9, while the number for the other four stages remained unchanged (*N* = 10).

### 2.6. Expanding Application to Developmentally Slow clk-1(e2519) Mutant Embryos

At first, we compared the duration of each embryonic stage between the *control(RNAi)* group and the WT group. No significant differences were detected across the five epidermal stages (see Appendix A). Unlike the three genes tested in the RNAi group, it has been shown that *clk-1* regulates developmental timing across the entire life cycle of the worm [34,35]. After manually correcting the misclassifications, the average duration of each *clk-1* embryonic stage was evaluated. The results indicate that, compared to the WT, the *clk-1(e2519)* mutant exhibited marked developmental delays in five epidermal stages, including the dorsal intercalation stage (delayed by 76.8%; *p* < 0.001), ventral enclosure stage (delayed by 265.2%; *p* < 0.01), rotation stage (delayed by 424.1%; *p* < 0.05), 1.5-fold stage (delayed by 236.4%; *p* < 0.001), and 2-fold stage (delayed by 171.4%; *p* < 0.001) (Table 6; Figure 7). We also noticed that the transition between dorsal intercalation and ventral enclosure was significantly longer in *clk-1(e2519)* mutants (*p* < 0.05) (see Appendix A).

## 3. Discussion

In this study, we proposed a deep learning-based approach for analyzing *C. elegans* epidermal morphogenesis. Unlike traditional approaches that focus on defective phenotypes, our approach for gene function analysis focused on the time required for completion of each developmental stage.

### 3.1. Deep Learning-Based Diagnostic Tool for Temporal Analysis

We implemented a new diagnostic approach for epidermal morphogenesis using two classic deep learning models. First, ResU-Net was employed to isolate developmentally complete embryos for subsequent detection. Next, ResNet was incorporated into the pipeline to determine the embryonic stage from the time-lapse images. What sets our approach apart is that we adopted a modular two-step pipeline instead of using an integrated model or directly applying a classification network. In the first step, ResU-Net is used to generate pixel-accurate masks, which reduces time required for annotation and allows the classification stage to focus on meaningful areas, thereby improving prediction accuracy. Moreover, the modular design enhances the pipeline’s reusability and extensibility; for example, when analyzing early embryonic timelines, only the classification model needs retraining, while the pretrained ResU-Net can be reused.

For ResNet, the softmax function’s ability to assign probabilities to classification labels was used to dynamically visualize the embryonic development process and transitional periods. Furthermore, the UMAP projection of the predicted labels after training showed that the classified stages clustered sequentially along the correct embryonic developmental trajectory rather than being randomly distributed, indicating that the model had learned semantically meaningful features aligned with the temporal progression of morphogenesis.

### 3.2. Calculations of Time Required for Each Stage in control(RNAi) Group

In this study, we defined the strict 0-min time point at the 2-cell stage of the embryo. Time-lapse images were then captured every 5 min, starting from 150 min, until the completion of the 2-fold stage. Using time as the horizontal axis and the predicted probability of each stage as the vertical axis, we created a timeline from the dorsal intercalation stage to the 2-fold stage, with different colors representing each stage. By measuring the duration of different colored segments in the predicted timeline, we were able to estimate the duration of each embryonic stage.

Although the current approach occasionally misclassified individual images within the entire time-lapse dataset, the fixed epidermal development process made such errors easily identifiable. For example, misclassified images appeared as isolated points of a different color within a continuous segment on the predicted timeline, making them visually distinguishable. Therefore, when images were misclassified (i.e., appeared as a different stage within a segment of a single predicted color on the timeline) they were assigned to the corresponding embryonic stage for the purpose of calculating the duration of each stage.

During the model training phase, no specific constraints were imposed on embryo orientation. As a result, when applying the model to actual time-lapse data, the method could process images unrestrictedly as long as the entire embryo fit within the 256 × 256 pixels frame. In the *control(RNAi)* group, the standard error of the mean for each stage was within 2 min, indicating low variability in our measurements. However, the measurement of duration for the dorsal intercalation and ventral enclosure stages showed greater individual variability compared to later stages. This variability might be due to the embryos within the eggshell not aligning their ventral or dorsal side toward the microscope, or to some embryos being positioned at an angle to the eggshell. This could introduce inconsistency in the duration measurements during these stages. Further analyses with angle correction may improve measurement accuracy in future studies.

### 3.3. Trial to Temporal Analysis of the Gene Function in RNAi Knockdown Animals

This study employed a diagnostic approach that applies AI-image-based classification to analyze temporal patterns. Specifically, it divided epidermal morphogenesis into five stages and calculated the duration of each stage. Using this method, we detected developmental delays in three selected genes at specific embryonic stage, clarifying the stages of epidermal development to which each gene specifically contributed.

In the analysis of *leo-1(RNAi)* animals, developmental delays were observed in three early embryonic stages: dorsal intercalation: (delayed by 20.0%), ventral enclosure (delayed by 56.5%), rotation (delayed by 44.1%). No delays were detected at the 1.5-fold and 2-fold stages. This suggests that *leo-1* was primarily involved in early epidermal development and did not contribute substantially to later stages. *leo-1*, as a component of the PAF1C complex, collaborated with the other four components to regulate cell migration and cell positioning, which are critical for epidermal development [31]. In the current study, the delays observed from the dorsal intercalation to the rotation stage in *leo-1(RNAi)* animals likely resulted from reduced LEO-1 expression in early epidermal development, impairing the timely progression of normal cell migration and positioning. Since no delays were observed at the 1.5-fold and 2-fold stages, it is hypothesized that the reduced *leo-1* expression permitted cell positioning to proceed with delays but without major disruption. Consequently, late-stage epidermal development could proceed on schedule. This further suggests that *leo-1* had weak or no contribution to later stages of epidermal development.

In the analysis of *ajm-1(RNAi)* animals, developmental delays were observed at two late embryonic stages: 1.5-fold (delayed by 31.4%) and 2-fold (delayed by 50.7%). No delays were detected from dorsal intercalation to the rotation stage. This suggests that *ajm-1* contributed primarily during the elongation phase of late epidermal development, with minimal involvement in earlier stages. *ajm-1* is essential for the integrity of epithelial junctions and embryonic elongation [32]. Our temporally detailed analysis is consistent with these findings. Although previous research reported developmental delays in *ajm-1* embryos [32], the reported timing spanned a broad period from enclosure to the 2-fold stage. Using our approach, we clarified that delays specifically occurred at the 1.5-fold and 2-fold stages, providing complementary insights into previous findings.

In the analysis of *tes-1(RNAi)* animals, developmental delays were observed only at the 2-fold stage (delayed by 93.1%). This suggests that *tes-1* played a specific role in embryonic elongation starting from the 2-fold stage, with minimal contribution to early epidermal development and the elongation process prior to the 2-fold stage. TES-1 localizes to junctions in a tension-dependent manner from the 2-fold stage, stabilizing the junctional actin cytoskeleton during embryonic morphogenesis [33]. Therefore, the observed 2-fold stage delay in *tes-1(RNAi)* animals likely reflects the critical role of TES-1 in maintaining cell–cell contacts necessary for subsequent embryonic elongation. Our findings confirm previous studies and provide new insights based on a different analytical approach.

### 3.4. Temporal Analysis Reveals That the Developmental Time Course Is Comparable Between control(RNAi) and WT Embryos

No significant differences were observed across the five epidermal developmental stages between the *control(RNAi)* and WT groups. Although this outcome is not unexpected, these findings provide evidence that differences in nutritional composition between HT115 and OP50 do not affect epidermal morphogenesis in *C. elegans*. Furthermore, these findings validate the accuracy of the method adopted in this study.

### 3.5. Extended Trial to Temporal Analysis of Developmentally Slow clk-1(e2519) Mutant Embryos

To further explore the potential applicability of this method, we analyzed *clk-1(e2519)* embryos. While mutations in *clk-1* have been reported to extend various aspects of developmental timing, including embryonic development, postembryonic development, self-brood size, and lifespan [34], their specific effects on epidermal morphogenesis during embryogenesis have not been well characterized.

In the *clk-1(e2519)* mutant embryos, significant developmental delays were observed across all five epidermal stages. This suggests that epidermal morphogenesis is encompassed within the developmental progression regulated by the *clk-1* gene. Grad-CAM interpretability analysis indicates that the AI may be misled by the two distinct bends in the 1.5-fold stage images of *clk-1* caused by premature muscle-twitching. This premature muscle-twitching is likely caused by a temporal mismatch between muscle cell differentiation and the morphogenesis process induced by *clk-1* dysfunction. These results suggest that *clk-1* dysfunction may cause not only overall developmental delay but also cause mismatch between muscle cell differentiation and morphogenesis. We speculate that our AI-based temporal analysis method has advantages in analyzing the late morphogenesis stage. Therefore, we propose that our method complements existing tools, such as StarryNite and AceTree, to achieve a more comprehensive understanding of the genes involved in the regulation of morphogenesis.

### 3.6. Contributions and Limitations of the Current Approach

The proposed approach successfully diagnoses the specific developmental stages to which the three RNAi-treated genes contributed to embryonic epidermal development. Using this approach, genes identified through a genome-wide RNAi screen can be further analyzed to determine their specific contributions to developmental progression. This is particularly relevant for genes where development timing, as demonstrated with *tes-1(RNAi)* in this study, needs to be considered. By applying this approach to a secondary screening using *C. elegans* homologs of genes associated with epidermal-related diseases or developmental delay in other model organisms, it could serve as a complementary analytical approach to existing methods.

The current approach holds potential for applications requiring high-throughput analysis, such as worm-based drug or agrochemical screening. In this study, an NVIDIA RTX 3060 GPU was used, which processed approximately 50 sequential DIC images to generate a developmental timeline required less than 3 s, corresponding to a throughput of approximately 18 frames per second (fps) or 120 embryos per hour on a single GPU. Coupling this computational speed with a microscope capable of automatically acquiring time-lapse images of large numbers of *C. elegans* embryos would enable high-throughput comparative analyses. Furthermore, this approach holds potential for detecting certain developmental abnormality related diseases that arise during embryogenesis but remain undetected by conventional methods. In contrast to traditional studies that primarily focus on postnatal pathogenesis [36], our method emphasizes early screening, when subtle developmental defects occur.

However, the presented approach still suffers from a few limitations. During the transition periods, the model exhibits instability, as two distinct patterns were observed. This may be because transition periods display characteristics of mixed stages, and the model has not been specifically trained to recognize them, leading to uncertainty in predictions. During non-transition periods, to minimize potential human bias, we only corrected isolated misclassifications while excluding consecutive misclassifications. Therefore, a key challenge for further model optimization remains the incorporation of algorithms that account for temporal continuity in the predicted timeline, thereby addressing and preventing consecutive misclassifications.

In addition, increasing the diversity and size of the training dataset—for example, by incorporating embryo images collected at different time points, from different research institutes, public datasets, or acquired using different types of microscopes—would further enhance the model’s robustness and generalizability.

Another limitation is that the current approach produced several unstable timelines when predicting time-lapse data from the RNAi group and *clk-1* group. We suspect that this instability arose because the model detected subtle defects in the embryos following loss of gene function that are often imperceptible to the human eye. Exploring how deep learning can further leverage its ability to discern these subtle defects and uncover their underlying causes could be intriguing for future research.

## 4. Materials and Methods

### 4.1. Caenorhabditis elegans Strains

*C. elegans* strains used in this study were derived from the wild-type Bristol strain N2 [37]. CB4876: *clk-1(e2519)* and SU93: *jcls1 [ajm-1::GFP]* was obtained from the Caenorhabditis Genetics Center (CGC). SU93 was used to characterize various stages of epidermal development but was not used for training and timeline prediction experiments. All strains were cultured at 20 °C on Nematode Growth Medium plates with *Escherichia coli* OP50 as the food source.

### 4.2. RNA Interference Assay

The RNAi assay was performed as previously described [38]. The *leo-1(RNAi)* cells were a gift from a previous study [39]. To construct the RNAi vectors, total RNA was extracted from *C. elegans* using the RNeasy Mini Kit (QIAGEN, Hilden, Germany) and RNase-Free DNase Set (QIAGEN, Hilden, Germany) to eliminate genomic DNA contamination. Complementary DNA (cDNA) was subsequently synthesized using the PrimeScript™ 1st Strand cDNA Synthesis Kit (Takara, Kusatsu, Japan). Full-length *tes-1*, *elt-1*, *lin-26*, and *let-413* cDNA, and 978bp *ajm-1* (1st–978th coding region) were amplified from this library and inserted into the L4440 vector. The vector was then transformed into *E. coli* HT115 (DE3) cells. The *control(RNAi)* group utilized the L4440 vector that was transformed into HT115 cells without any cDNA insertion. To ensure that worms subjected to RNAi were stage-synchronized, L1 worms were first transferred to a new plate with OP50 and cultured at 20 °C for 24 h. Next, two to three healthy worms were randomly selected and transferred to RNAi feeding plated and cultured at 20 °C for 24 h with HT115.

### 4.3. Epidermal Morphogenesis Stages

The epidermal morphogenesis of *C. elegans* embryos was divided into six distinct stages (“before intercalation”, “dorsal intercalation”, “ventral enclosure”, “rotation”, “1.5-fold”, and “2-fold”) from the mid-embryo gastrulation stage to the early elongation phase. Before collecting the training data, the specific characteristics of these stages were identified using strain SU93: *jcls1 [ajm-1::GFP]* (see Appendix A).

#### 4.3.1. Before Intercalation

In this stage, the cells on the dorsal side of the embryo had not yet specialized into epidermal cells; therefore, many undifferentiated cells were observed (see Appendix A). The correlation between this stage and epidermal development is weak. Although this stage is included in the timeline prediction, it is excluded from the subsequent comparative analysis of the time required for each embryonic stage due to its weak relevance.

#### 4.3.2. Dorsal Intercalation

By adjusting the focus, the insertion of epidermal cells on the dorsal side was observed in this stage [2,40] (see Appendix A).

#### 4.3.3. Ventral Enclosure

By adjusting the focus, the ventral pocket on the ventral side of the embryo was observed [2] (see Appendix A).

#### 4.3.4. Rotation

In most research, this period of embryogenesis is referred to as the “comma” stage owing to the comma-like shape of the embryo. However, after epidermal cells enclose the entire posterior portion of the embryo, an imbalance along the anteroposterior axis causes the embryo to rotate within the egg [2]. Therefore, we abandoned the use of the term “comma” for this stage and used “rotation” instead. This terminology aligns with the focus on the dynamic changes in the embryo.

#### 4.3.5. 1.5-Fold and 2-Fold

In these two stages, the elongation of the embryo was quantified by observing its fold length; if the tail had elongated to align with the head, the embryo was considered to be at the 2-fold stage; otherwise, it was classified as 1.5-fold.

The embryo is considered to have completed the 2-fold stage when the worm’s head had overlapped the tail due to elongation, or conversely, if the tail has overlapped the head. The collection of time-lapse images is concluded once this stage is observed.

### 4.4. Microscope and Image Acquisition

A DIC Olympus BX63 microscope with a UPlanXApo X40 NA 0.95 objective lens was used to capture images of embryos (Olympus, Tokyo, Japan). The microscope system was operated using in-house CellSens Dimension 3.1 software (Olympus, Tokyo, Japan). Images were saved in the JPEG format at a resolution of 256 × 256 pixels. Automatic exposure was set for each image acquisition. The experimental environmental temperature was maintained at 23 °C.

The worms were dissected onto a coverslip with 2 μL of M9 buffer and mounted on glass slides with 3% agarose pads. All images were taken with the DIC microscope focused on the clearest dimension of the egg; however, no specific requirements were placed on embryo orientation (e.g., front, back, or angled) as long as the entire embryo fitted within the 256 × 256-pixel frame.

Time-lapse imaging was used to predict the developmental timeline of embryos and was not included in the training of the deep learning models. First, a Vaseline ring was applied around the coverslip to retain moisture facilitating long-term observation. To ensure accuracy, the timeline was initialized (0 min) after the first cleavage [41]. When the AB cell attained round shape and the P1 cell was compressed into a crescent shape (see Appendix A), thus minimizing timing errors. Images were collected at 5-min intervals using the 40× objective lens, with an axial (z) resolution of 0.8 μm. At each time point, the clearest image of the egg was selected from a series of images captured along the z-axis. Data collection began at 150 min focusing on the epidermal development phase and to avoid unnecessary computational overhead. Imaging was continued until the end of the 2-fold stage or until 500 min for embryos exhibiting developmental defects, ensuring sufficient observation of both defective and delayed development.

### 4.5. Models Training and Evaluation

#### 4.5.1. ResU-Net

The segmentation model used in this study was a modified U-Net [28,29] architecture, referred to here as a ResU-Net, which integrates residual connections into both the encoder and decoder blocks, thereby improving gradient flow and training stability. The model takes single-channel grayscale images of size 256 × 256 pixels as input. Each encoder and decoder stage consists of two residual blocks, with each block comprising two 3 × 3 convolutional layers, followed by batch normalization and ReLU activation. Max pooling layers were employed for spatial downsampling, and transposed convolutions were used for upsampling. Skip connections were applied between corresponding encoder and decoder layers to preserve spatial information. The number of feature maps doubled after each downsampling and halved during upsampling. A final 1 × 1 convolution layer was used to project the output to a single-channel segmentation mask (Figure 8).

ResU-Net was implemented in PyTorch (v2.0.0) and trained for 80 epochs with a batch size of 20. The Adam optimizer was used with a learning rate of 0.002 and a dropout rate of 0.2. As the task was binary segmentation (embryo vs. background), the loss function used was BCEWithLogitsLoss, which combines a sigmoid layer and the binary cross-entropy loss in a numerically stable formulation.(1)$$L_{BCE}=-\frac{1}{N}\sum_{i=1}^{N}\left[y_{i}\;\cdot \log\left(\sigma \left(x_{i}\right)\right)+(1-y_{i})\cdot log(1-\sigma (x_{i}))\right]$$(2)$$\sigma \left(x\right)=\frac{1}{1+e^{-x}}$$
where N represents the number of samples; $y_{i}$ represents the ith true tensor of the sample; $x_{i}$ represents the ith predicted tensor of sample.

All input images were converted to single-channel grayscale images (via torchvision.transforms, torchvision v0.15.0). The corresponding ground truth segmentation masks were annotated using LabelMe (v5.1.1), where the embryo’s outer contours (eggshell) were manually outlined to generate binary masks. The dataset was randomly split into training and validation sets with an 80:20 ratio. During training, model performance was evaluated using the Intersection over Union (IoU) metric. The best-performing model was selected based on the highest validation IoU.(3)$$IoU=\frac{\sum_{i=1}^{N}\hat{y_{i}}y_{i}\;}{\sum_{i=1}^{N}\hat{y_{i}}+y_{i}-\hat{y_{i}}y_{i}\;}$$(4)$$\hat{y_{i}}=\left\{\begin{array}{c} 1,\;\;if\;x_{i}\;\geq 0\\ \;\;0,\;\;otherwise \end{array}\right.$$
where N represents the number of samples; $y_{i}$ represents the ith true tensor of the sample; $\hat{yi}\;\in \;\left\{0\right.,\left.1\right\}$ represents the predicted binary mask obtained by thresholding the model raw output, the binarization corresponds to applying a threshold of 0.5 to the sigmoid output $\sigma \left(x\right)$.

#### 4.5.2. ResNet

The classification model was based on ResNet-18 [30] from torchvison (v0.15.0). The input images were 256 × 256-pixel single-channel grayscale images obtained from the ResU-Net. The classification task involved six distinct developmental stages of the embryo, with 400 labeled images per stage, yielding a total of 2400 annotated images. Each image corresponded to a precisely staged embryo (Figure 9).

The model was trained from scratch without using any pretrained weights. Training was performed using the CrossEntropyLoss function and optimized with the Adam optimizer. A step learning rate scheduler (StepLR) was applied, reducing the learning rate by a factor of 0.5 every 10 epochs to facilitate convergence. Training was conducted for 80 epochs. Model performance was evaluated using classification accuracy on the validation set.

In addition, considering the relatively small size of the dataset, we performed a five-fold cross-validation using the combined dataset (*N* = 400 images per class). In each fold, 80.0% of the data were used for training and 20.0% for validation. The model architecture and training hyperparameters were identical to those in the main experiment.

### 4.6. Timeline

To obtain a timeline of embryonic epidermal development, the two deep learning architectures were integrated. First, the segmentation network processed the selected time-lapse images. The output of the segmentation network assigned a tensor value of 1 to the embryo (within the egg) and 0 to the background. By multiplying the predicted tensor (segmented result) with the input tensor (original image), a processed image was generated, retaining only the pixel values of the embryonic part (Figure 10). Because only the embryonic part of the processed image retained the pixel values, the classification network in the next phase focused on the features of the embryonic region. This modular design allows the segmentation module to be reused in future analyses without retraining.

The processed images were then passed to the classification network, which predicted the probability of each labeled developmental stage from the softmax layer. Finally, the predicted probabilities were plotted on the *Y*-axis, and the embryonic development time was plotted on the *X*-axis. This produced a dynamic timeline where embryonic epidermal development was visualized in chronological order (Figure 11).(5)$$softmax\left(x_{i}\right)=\frac{exp\left(x_{i}\right)}{\sum_{j}exp\left(x_{j}\right)}$$
where $x_{i}$ represents the parameters of ith avgpool; j denotes the number of labels.

### 4.7. Image Interpretability Analysis

#### 4.7.1. Grad-CAM

Gradient-weighted class activation mapping (Grad-CAM) was used to verify the accuracy of image predictions. Briefly, this technique verified that the deep learning network had focused on the relevant regions of the embryo during prediction. Grad-CAM highlights the important features in an image by analyzing the gradients flowing into the last convolutional layer, thus generating an activation map that identifies the most discriminative areas contributing to the classification decision.

#### 4.7.2. UMAP

UMAP was used to obtain the distribution of semantic features. The ResNet model was first trained on the image dataset, and semantic features were extracted from the output of the AvgPool layer of each validation image using the create_feature_extractor function from torchvision (v0.15.0). These high-dimensional features were then projected into a two-dimensional space using UMAP with Euclidean distance as the metric. The resulting UMAP embedding provided an interpretable visualization of how images were distributed in the feature space, reflecting the temporal progression learned by the model.

## 5. Conclusions

In this study, a deep learning-based diagnostic approach was developed to evaluate and analyze the temporal perspective of *C. elegans* epidermal morphogenesis. We aimed to address the gap generated in traditional methods, which often overlook the importance of temporal aspects during development, particularly in cases where the inactivation of certain genes does not result in apparent phenotypic defects. Temporal analysis indicated stage-specific delays in embryonic development, emphasizing the importance of developmental timing as a critical parameter when analyzing epidermal morphogenesis and providing a new perspective for studying developmental processes.

Despite some limitations, the current study undoubtedly augments traditional methods and plays an important role in epidermal morphogenesis research. Moreover, it offers unique temporal insights and provides an accessible and cost-effective tool for studying morphogenesis and developmental timing. This strategy may also have broad applications in other image-based studies of morphogenesis, such as embryogenesis in various organisms or early diagnosis of developmental disorders that are not readily apparent during embryogenesis.

## Figures and Tables

**Figure 1 ijms-26-10802-f001:**
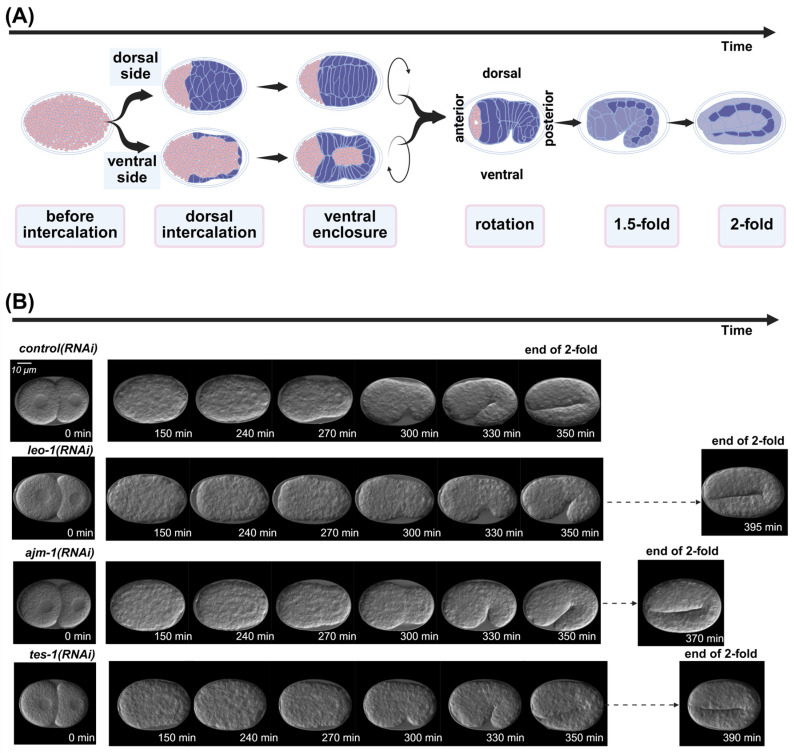
Key events in *Caenorhabditis elegans* embryonic epidermal development process. (**A**) Six stages of embryonic epidermal development in *C. elegans*, which were predicted using deep learning models. (**B**) Embryos exhibiting developmental impacts temporally following RNA interference treatment. Scale bar: 10 μm. All images were processed by the ResU-Net model trained in this study.

**Figure 2 ijms-26-10802-f002:**
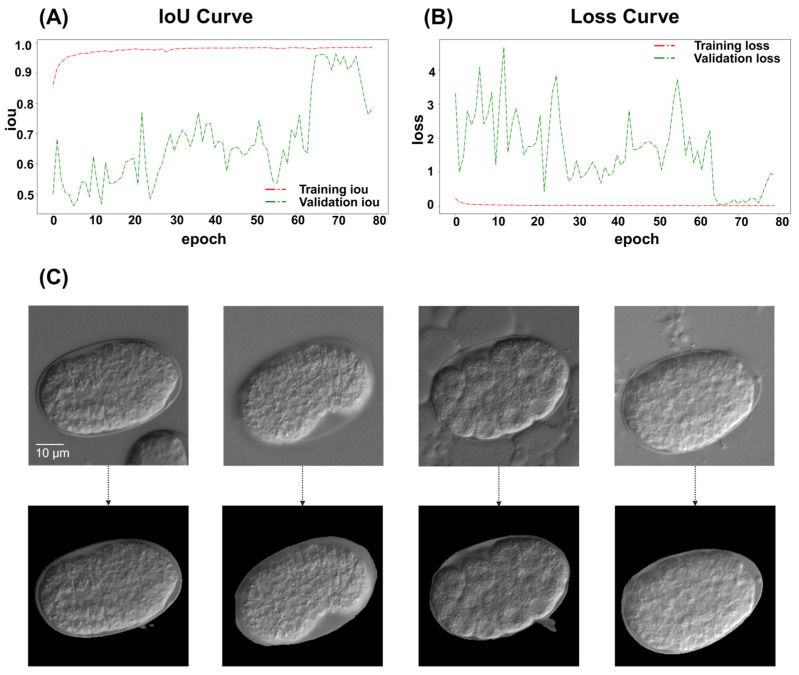
ResU-Net performance diagram. (**A**) ResU-Net IoU results diagram. The IoU scores are used to evaluate accuracy. The green line represents the IoU score for the validation set, while the red line indicates the IoU score for the training set. (**B**) ResU-Net loss diagram. The green line represents the loss for the validation set, while the red line indicates the loss for the training set. (**C**) ResU-Net also performs well in deliberately prepared images with severe noise. From left to right, images show: mixed with other embryos in the frame, focus issues, presence of bubbles around the embryo, and presence of tissue debris around the embryo. Scale bar: 10 μm. The top row shows the original images, whereas the bottom row displays the processed results.

**Figure 3 ijms-26-10802-f003:**
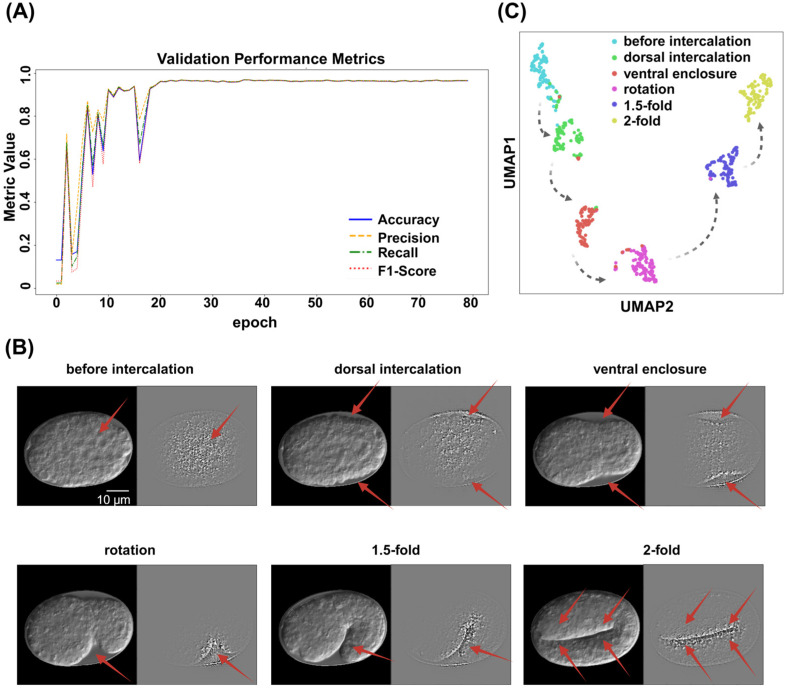
ResNet performance diagram. (**A**) ResNet validation performance diagram. Different colors of blue, orange, green, and red, represent “Accuracy”, ”Precision”, ”Recall”, and “F1-Score”, respectively. (**B**) Gradient-weighted Class Activation Mapping (Grad-CAM) reveals that the image classification model can accurately capture the key semantic features of each embryonic stage. On the left side of each subcategory is the embryo image after ResU-Net processing, whereas the right side displays the image following Grad-CAM analysis. Red arrows indicate areas with high weightings. Scale bar: 10 μm. (**C**) UMAP visualization of feature representations from the validation dataset using the ResNet model. Each point represents a single embryo image, colored according to its predicted developmental stage. Black lines indicate the correct developmental trajectory of *C. elegans*, illustrating the temporal order of embryogenesis.

**Figure 4 ijms-26-10802-f004:**
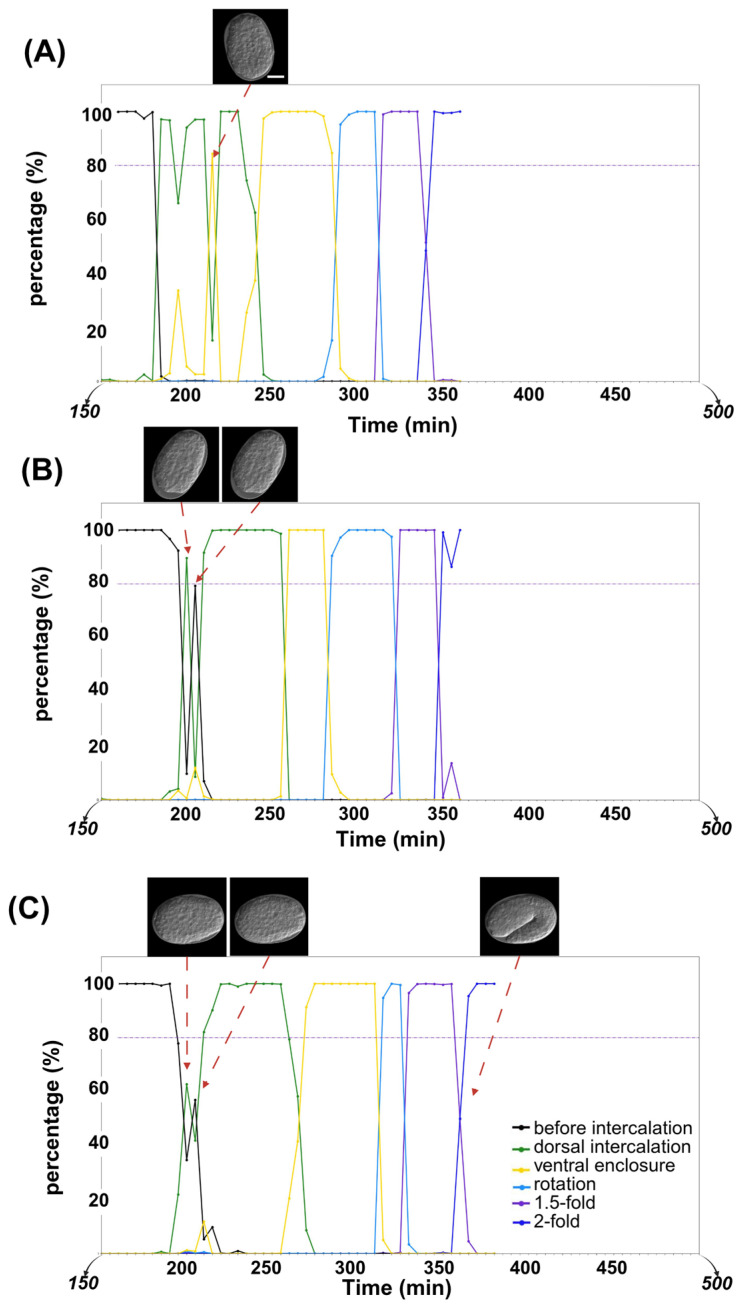
Screened timeline examples that show misclassification and the two observed patterns of transition periods. Scale bar: 10 μm. Different colors of black, green, yellow, sky blue, purple, and dark blue represent “before intercalation”, “dorsal intercalation”, “ventral enclosure”, “rotation”, “1.5-fold”, and “2-fold” stages, respectively. (**A**) Misclassification in the timeline. (**B**) Oscillating between high probabilities for two different stages during transition period. (**C**) Maintaining a low probability during transition period.

**Figure 5 ijms-26-10802-f005:**
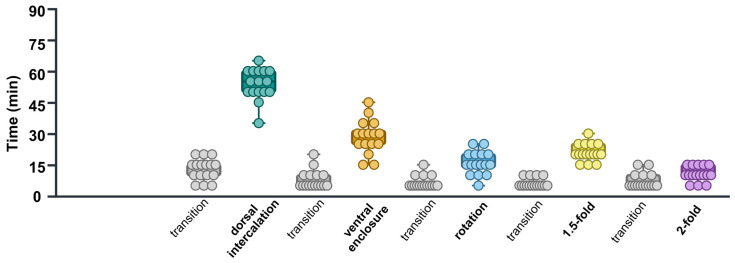
Time requirements for *control(RNAi)* embryonic development in each stage. Different colors of, green, orange, sky blue, yellow, purple, and grey represent “dorsal intercalation”, “ventral enclosure”, “rotation”, “1.5-fold”, and “2-fold” “transition” stages, respectively. Each dot represents the time required by a single embryo to undergo each specific embryonic stage.

**Figure 6 ijms-26-10802-f006:**
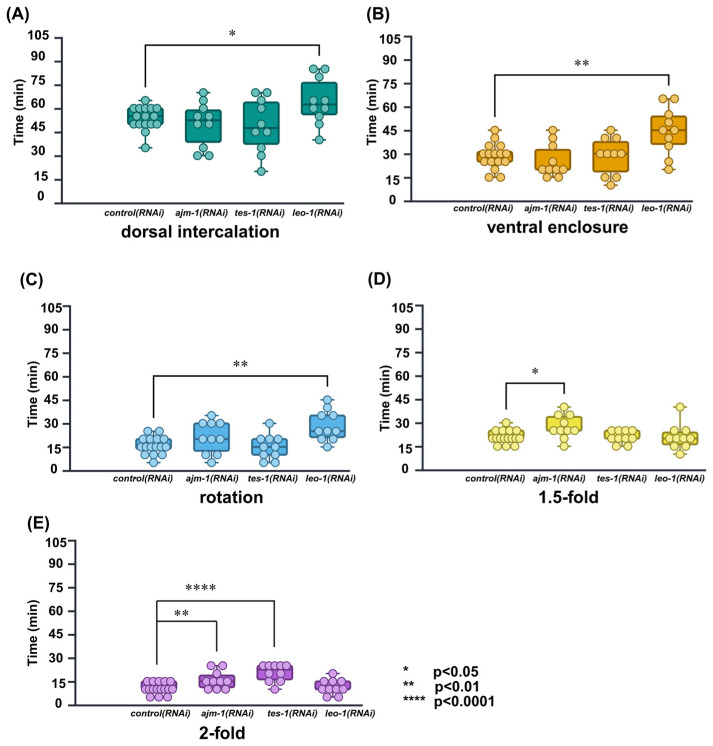
Calculated and compared duration of embryonic development in each stage and, successfully defined specific embryonic stages affected by RNAi treatment of *ajm-1*, *tes-1*, and *leo-1*. *ajm-1(RNAi)* was delayed in the 1.5-fold, 2-fold stages and, transition periods; *tes-1(RNAi)* was delayed in the 2-fold stage; *leo-1(RNAi)* was delayed in dorsal intercalation, ventral enclosure, and rotation stages. *control(RNAi)*: *N* = 16; *ajm-1(RNAi)*, *tes-1(RNAi)*, and *leo-1(RNAi)*: *N* = 10; * *p* < 0.05; ** *p* < 0.01; **** *p* < 0.0001; unpaired Student’s *t*-test. (**A**) dorsal intercalation. (**B**) ventral enclosure. (**C**) rotation. (**D**) 1.5-fold. (**E**) 2-fold. Each dot represents the time required by a single embryo to undergo each specific embryonic stage.

**Figure 7 ijms-26-10802-f007:**
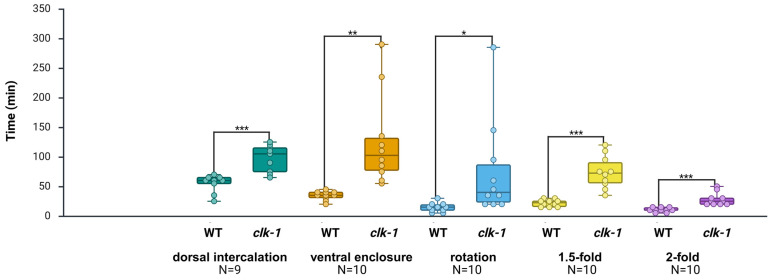
Calculated and compared duration of embryonic development at each stage in WT and *clk-1(e2519)* mutant embryos. The analysis successfully identified specific embryonic stages affected by *clk-1*. The *clk-1(e2519)* mutant exhibited severe developmental delays across all five epidermal stages examined. * *p* < 0.05; ** *p* < 0.01; *** *p* < 0.001; unpaired Student’s *t*-test. Each dot represents the duration of a single embryo for the corresponding developmental stage. Dorsal intercalation: (WT: *N* = 9; *clk-1*: *N* = 9); ventral enclosure: (WT: *N* = 10; *clk-1*: *N* = 10); rotation: (WT: *N* = 10; *clk-1*: *N* = 10); 1.5-fold: (WT: *N* = 10; *clk-1*: *N* = 10); 2-fold: (WT: *N* = 10; *clk-1*: *N* = 10).

**Figure 8 ijms-26-10802-f008:**
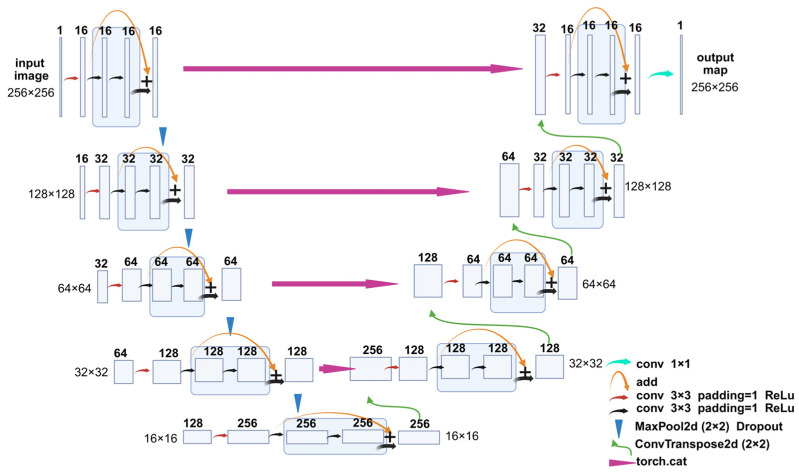
ResU-Net architecture. Residual blocks are incorporated into each encoder and decoder layer. The number of channels is indicated above each box. The tensor size is indicated on the side of box. Different arrows represent different operations, conv 1 × 1/conv 3 × 3: convolution with 1 × 1/3 × 3 kernel size; Add: element-wise addition; MaxPool2d: two-dimensional max pooling; ConvTranspose2d: two-dimensional transposed deconvolution; torch.cat: tensor concatenation.

**Figure 9 ijms-26-10802-f009:**
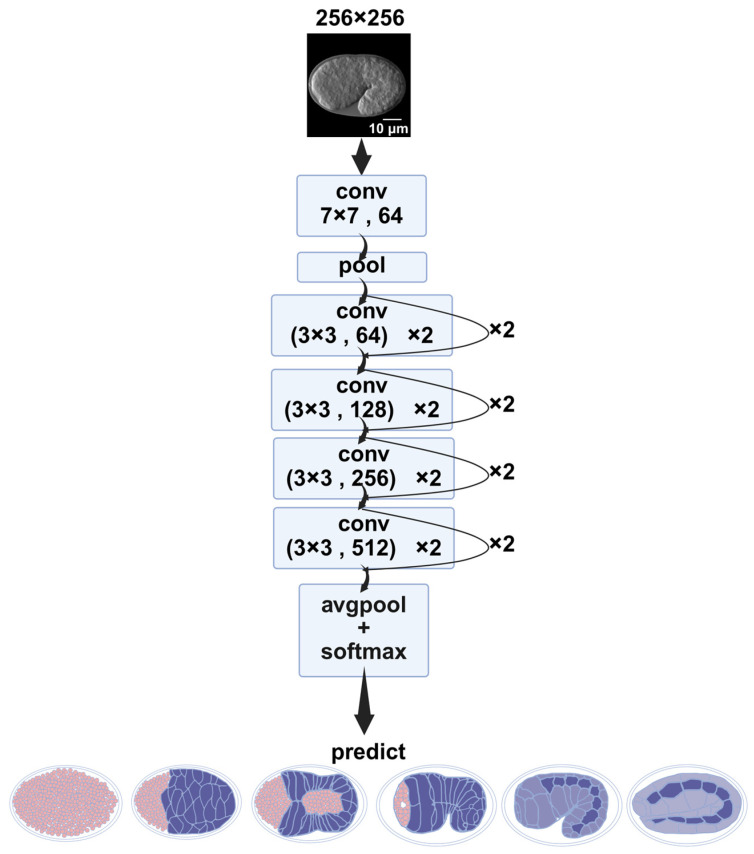
ResNet architecture. After the convolution operation in the residual blocks, softmax is applied to assign probability values to the images being predicted. Boxes represent different operations, conv, convolution layer; Pool, pooling layer; AvgPool, average pooling; the bottom shows the predicted labels from left to right: “before intercalation”; “dorsal intercalation”; “ventral enclosure”; “rotation”; “1.5-fold”; “2-fold”. Scale bar: 10 μm.

**Figure 10 ijms-26-10802-f010:**
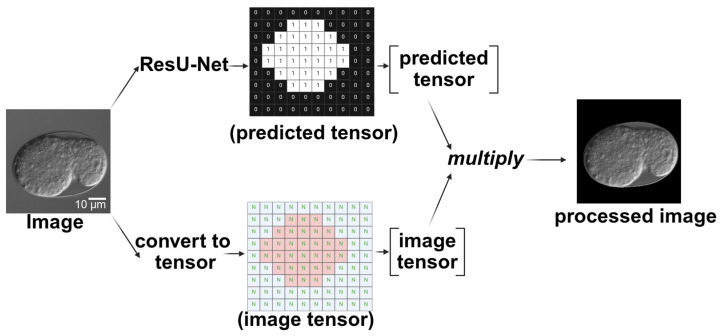
Schematic diagram of using ResU-Net processing to extract the embryonic region in the image. After passing through ResU-Net, a tensor containing only 0 s and 1 s is produced. This predicted tensor is then multiplied by the original image tensor to produce the processed image. Scale bar: 10 μm.

**Figure 11 ijms-26-10802-f011:**
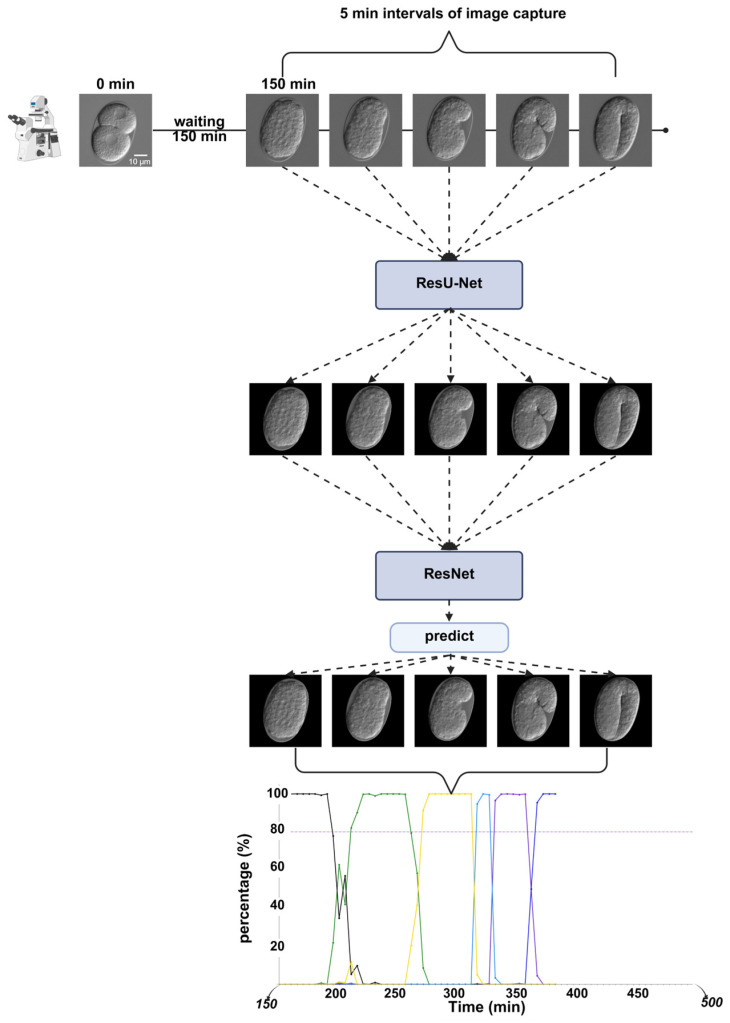
Key temporal analysis experimental workflow. The time-lapse image dataset is first processed by ResU-Net, followed by input into ResNet for embryonic stage prediction. Finally, the predicted timeline results are visualized. Scale bar: 10 μm.

**Table 1 ijms-26-10802-t001:** ResU-Net performance metrics.

Metric	Value
True Positive (TP)	519,293
False Positive (FP)	14,760
True Negative (TN)	714,840
False Negative (FN)	4483
Sensitivity (TPR)	99.1%
Specificity (TNR)	98.0%
Overall Accuracy	98.5%
Precision (PPV)	97.2%
F1-Score	98.2%
Intersection over Union (IoU)	96.4%

**Table 2 ijms-26-10802-t002:** ResNet performance metrics.

Metric	Value
Sensitivity (TPR)	96.9%
Specificity (TNR)	98.0%
Overall Accuracy	96.9%
Precision (PPV)	96.9%
F1-Score	96.8%

**Table 3 ijms-26-10802-t003:** Statistics of the timelines and number of images collected using different interference RNA.

RNAi	Number of Timelines	Timeline Number with Misclassification	Total Images	Images Number with Misclassification	Continuous Misclassification
*control(RNAi)*	16	1	681	0.2% (*n*= 1)	0
*leo-1(RNAi)*	10	2	541	0.4% (*n* = 2)	0
*ajm-1(RNAi)*	10	2	467	0.6% (*n* = 3)	0
*tes-1(RNAi)*	10	3	457	0.7% (*n* = 3)	0

**Table 4 ijms-26-10802-t004:** The average time required for each stage of epidermal development in RNAi group.

RNAi	Dorsal Intercalation (min)	Ventral Enclosure (min)	Rotation (min)	1.5-Fold (min)	2-Fold (min)
*control(RNAi)*	53.75 ± 1.85	28.43 ± 2.02	15.93 ± 1.38	20.93 ± 1.04	10.62 ± 0.89
*leo-1(RNAi)*	64.50 ± 4.74 *	44.50 ± 4.80 **	28.50 ± 3.08 **	21.00 ± 2.56	11.50 ± 1.50
*ajm-1(RNAi)*	50.50 ± 4.47	25.50 ± 3.37	21.00 ± 3.23	27.50 ± 2.39 *	16.00 ± 1.80 **
*tes-1(RNAi)*	49.50 ± 5.46	28.50 ± 3.73	15.00 ± 2.47	21.00 ± 1.25	20.50 ± 1.74 ****

* *p* < 0.05; ** *p* < 0.01; **** *p* < 0.0001.

**Table 5 ijms-26-10802-t005:** Statistics of the timelines and number of images collected using mutant embryos.

Strains	Number of Timelines	Timeline Number with Misclassification	Total Images	Images Number with Misclassification	Continuous Misclassification
WT	10	2	383	0.5% (*n* = 2)	1
*clk-1(e2519)*	10	7	1000	2.3% (*n* = 23)	1

**Table 6 ijms-26-10802-t006:** The average time required for each stage of epidermal development in *clk-1(e2519)*.

Strains	Dorsal Intercalation (min)	Ventral Enclosure (min)	Rotation (min)	1.5-fold (min)	2-fold (min)
WT	55.00 ± 5.00	34.50 ± 2.41	14.5 ± 2.41	22.00 ± 1.86	10.50 ± 1.17
*clk-1(e2519)*	97.22 ± 7.60 ***	126.00 ± 24.45 **	76.00 ± 26.41 *	74.00 ± 8.69 ***	28.50 ± 3.42 ***

* *p* < 0.05; ** *p* < 0.01; *** *p* < 0.001.

## Data Availability

All programs and data used in this study are publicly available on GitHub https://github.com/Fangzheng-py/worm_0.git (accessed on 25 April 2025).

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
