# Peer review of "Temporal Analysis of Embryonic Epidermal Morphogenesis in Caenorhabditis elegans"

_ijms, 2025, doi:10.3390/ijms262110802_

Round 1

Reviewer 1 Report

Comments and Suggestions for Authors

  The authors present a two–step deep-learning pipeline that first denoises/segments C. elegans embryos with a ResU-Net and subsequently assigns each frame of a DIC time-lapse movie to one of six epidermal stages with a ResNet-18 classifier. By converting the per-frame soft-max scores into a coloured timeline, they are able to measure the duration of individual morphogenetic stages and uncover stage-specific delays after RNAi of ajm-1, tes-1 and leo-1.  As most image-based screens still rely on overt phenotypes, the manuscript offers a conceptually new and potentially powerful way to score more subtle temporal defects.

Merit

  1. Conceptual novelty: the focus on developmental timing rather than morphology is timely and fills a methodological gap in C. elegans embryology.
  2. Technical soundness: both networks are trained from scratch; performance metrics (IoU ≈ 96 %, Accuracy ≈ 97 %) are solid and the Grad-CAM/UMAP analyses nicely document that the classifier has learnt biologically meaningful features.
  3. Biological validation: the pipeline correctly recapitulates published roles of some genes, demonstrating that the method is biologically informative even when no obvious shape defect is present.
  4. Re-usability: code and data have been released on GitHub, facilitating reproduction and re-use by the community.

Points that require clarification

  1. The authors opted for a separate segmentation (ResU-Net) and classification (ResNet) step. Why is an integrated detector such as YOLOv7, CenterNet2 or a Mask-RCNN variant was not employed. The authors should justify their design choice explicitly (e.g. need for pixel-accurate masks, small training set, better interpretability, domain transfer to fluorescent channels, etc.) and, if possible, provide a brief quantitative comparison or at least a discussion of expected trade-offs (speed, memory, annotation cost, ability to capture partial embryos, scalability to high-throughput setups).
  2. Dataset size, cross-validation and external test set: The training set contains only 400 images per class. A five-fold cross-validation or, ideally, a test on data acquired on a different day/microscope would strengthen claims of robustness and generalisation.
  3. Manual relabelling of scattered points introduces an element of human bias. Please state how long the correction takes per movie and whether results change if no correction is applied (could be shown in a supplementary figure).
  4. The method is indeed attractive for screening lead drugs or agrochemical leads. A paragraph in the Discussion describing how many embryos per condition could realistically be analysed, the current processing speed (fps or movies/hour on a single GPU) and potential automation of worm picking/mounting would help readers judge feasibility for large-scale screens.

Minor text issues

  1. Lines 89–103 belong to the Discussion rather than the Introduction.
  2. Lines 122–132, 195–196 and 220: when a figure is called out in the main text, the internal caption “Figure x” is not needed.
  3. In Figure 6, colours for the stages are not colour-blind friendly (green/red). Consider colour-blind-safe palettes.

Author Response

Responses to the comments of Reviewer #1

Comment 1:

The authors opted for a separate segmentation (ResU-Net) and classification (ResNet) step. Why is an integrated detector such as YOLOv7, CenterNet2 or a Mask-RCNN variant was not employed. The authors should justify their design choice explicitly (e.g. need for pixel-accurate masks, small training set, better interpretability, domain transfer to fluorescent channels, etc.) and, if possible, provide a brief quantitative comparison or at least a discussion of expected trade-offs (speed, memory, annotation cost, ability to capture partial embryos, scalability to high-throughput setups).

Response 1:

We agree with this comment.

We have added more detailed discussion in the revised manuscript to clarify the rationale for this design choice.

(p. 11, lines 297-302; p.12, lines 303-306)

Comment 2:

Dataset size, cross-validation and external test set: The training set contains only 400 images per class. A five-fold cross-validation or, ideally, a test on data acquired on a different day/microscope would strengthen claims of robustness and generalization.

Response 2:

We greatly appreciate the reviewer for pointing out this issue.

We added a five-fold cross-validation result to evaluate the robustness of the ResNet model

(p. 5, lines 137-139; p. 18 lines 576-579)

In addition, we have added text discussion that improving the model’s robustness and generalizability will be crucial for future work.

(p. 14 lines 441-444)

Comment 3:

Manual relabeling of scattered points introduces an element of human bias. Please state how long the correction takes per movie and whether results change if no correction is applied (could be shown in a supplementary figure).

Response 3:

We thank the reviewer for pointing out this issue.

Since misclassifications can be rapidly identified based on the temporal progression of embryonic development, the manual correction takes less than 1 min per movie.

Following the reviewer’s suggestion, we recalculated the average duration of each embryonic stage without any manual correction.

(p. 9 lines 225-229)

Comment 4:

The method is indeed attractive for screening lead drugs or agrochemical leads. A paragraph in the Discussion describing how many embryos per condition could realistically be analysed, the current processing speed (fps or movies/hour on a single GPU) and potential automation of worm picking/mounting would help readers judge feasibility for large-scale screens.

Response 4:

We thank the reviewer for this helpful suggestion.

We have added the text in the “Discussion” section to highlight the potential of our approach for high-throughput applications, such as worm-based drug or agrochemical screening.

(p. 14, lines 420-427)

Comment 5:

Lines 89–103 belong to the Discussion rather than the Introduction.

Response 5:

We agree with this comment.

We have excluded the corresponding text from the “Introduction”.

Comment 6:

Lines 122–132, 195–196 and 220: when a figure is called out in the main text, the internal caption “Figure x” is not needed.

Response 6:

We sincerely apologize for several citation errors occurring the submitted manuscript. These issues may have arisen during the submission process, and we greatly regret any inconvenience they may have caused you during the reviewing. We have carefully reviewed the manuscript and corrected all the apparent citation errors in the revised version.

Comment 7:

In Figure 6, colors for the stages are not color-blind friendly (green/red). Consider color-blind-safe palettes.

Response 7:

We agree with this comment.

We have updated all figures accordingly, replacing colors with a color-blind–friendly palette.

(p. 8 Figure 5; p.9 Figure 6; p.11 Figure 7)

Reviewer 2 Report

Comments and Suggestions for Authors

Summary:

In the presented study, researchers described deep learning–based image analysis

pipeline that combines ResU-Net for noise reduction and ResNet for embryonic stage

prediction. They applied these two previously developed deep learning models to images

collected via time-lapse diKerential interference contrast (DIC) microscopy. Authors

claimed that their pipeline could accurately predict five diKerent developmental stages of

C. elegans embryos; just with an input of DIC images collected over a time-period.

Overall, it is a below average tool with a very narrow potential applicability. Additionally,

there are existing programs like “StarryNite” and “AceTree” to explore embryonic

development, including lineage tracing.

To be considered for publication, authors need to demonstrate their pipeline tool’s

potential application using available genetic mutants, for instance clk-1, clk-2, and clk-3

mutants!

Author Response

Responses to the comments of Reviewer #2

Comment 1:

It is a below average tool with a very narrow potential applicability. Additionally, there are existing programs like “StarryNite” and “AceTree” to explore embryonic development, including lineage tracing.

Response 1:

We thank the reviewer for pointing out this critical issue.

We have added a paragraph in the “Introduction” and expanded the discussion in the “Discussion” section to clarify that our method is intended to complement existing methods such as StarryNite and AceTree.

(p. 6, lines 81-87; p.14, lines 403-409)

Comment 2:

To be considered for publication, authors need to demonstrate their pipeline tool’s potential application using available genetic mutants, for instance clk-1, clk-2, and clk-3 mutants!

Response 2:

We thank the reviewer for the helpful suggestion.

To expand the potential application of our method, we additionally performed a temporal analysis using the developmentally delayed clk-1 mutant.

We have added the corresponding text in the “Results” and “Discussion” in the revised manuscript.

(p. 10, lines 244-279; p. 11, lines 280-289; p. 13, lines 383-400; p.14, lines 401-409)

Round 2

Reviewer 2 Report

Comments and Suggestions for Authors

I am satisfied with the revision. Only figure 6 needs formatting correction. Congratulations!

Author Response

Responses to the comments of Reviewer #2

Comment 1:

Only figure 6 needs formatting correction.

Response 1:

We thank the reviewer for pointing out this critical issue.

We revised Figure 6 and its legend to ensure consistent panel formatting (The positions of panels (A), (B), (C), (D), and (E) were changed. In the legend, (a), (b), (c), (d), and (e) were changed to (A), (B), (C), (D), and (E)). (p. 9, lines 239-240)